# The Effect and Cost-Effectiveness of Offering a Combined Lifestyle Intervention for the Prevention of Cardiovascular Disease in Primary Care: Results of the Healthy Heart Stepped-Wedge Trial

**DOI:** 10.3390/ijerph20065040

**Published:** 2023-03-13

**Authors:** Emma A. Nieuwenhuijse, Rimke C. Vos, Wilbert B. van den Hout, Jeroen N. Struijs, Sanne M. Verkleij, Karin Busch, Mattijs E. Numans, Tobias N. Bonten

**Affiliations:** 1Health Campus the Hague, Leiden University Medical Center, 2511 DP The Hague, The Netherlands; 2Department of Medical Decision Making, Leiden University Medical Center, 2333 ZA Leiden, The Netherlands; 3Department of Quality of Care and Health Economics, National Institute for Public Health and the Environment, 3720 MA Bilthoven, The Netherlands; 4Hadoks Chronische Zorg BV, 2517 JK The Hague, The Netherlands; 5Department of Public Health and Primary Care, Leiden University Medical Center, 2333 ZA Leiden, The Netherlands

**Keywords:** primary practice, preventative health care, cardiovascular risk, lifestyle factors, cost effectiveness

## Abstract

Objective: To evaluate the effectiveness and cost-effectiveness of offering the combined lifestyle programme “Healthy Heart”, addressing overweight, diet, physical activity, smoking and alcohol, to improve lifestyle behaviour and reduce cardiovascular risk. Design: A practice-based non-randomised stepped-wedge cluster trial with two-year follow-up. Outcomes were obtained via questionnaires and routine care data. A cost–utility analysis was performed. During the intervention period, “Healthy Heart” was offered during regular cardiovascular risk management consultations in primary care in The Hague, The Netherlands. The period prior to the intervention period served as the control period. Results: In total, 511 participants (control) and 276 (intervention) with a high cardiovascular risk were included (overall mean ± SD age 65.0 ± 9.6; women: 56%). During the intervention period, 40 persons (15%) participated in the Healthy Heart programme. Adjusted outcomes did not differ between the control and intervention period after 3–6 months and 12–24 months. Intervention versus control (95% CI) 3–6 months: weight: β −0.5 (−1.08–0.05); SBP β 0.15 (−2.70–2.99); LDL-cholesterol β 0.07 (−0.22–0.35); HDL-cholesterol β −0.03 (−0.10–0.05); physical activity β 38 (−97–171); diet β 0.95 (−0.93–2.83); alcohol OR 0.81 (0.44–1.49); quit smoking OR 2.54 (0.45–14.24). Results were similar for 12–24 months. Mean QALYs and mean costs of cardiovascular care were comparable over the full study period (mean difference (95% CI) QALYs: −0.10 (−0.20; 0.002); costs: EUR 106 (−80; 293)). Conclusions: For both the shorter (3–6 months) and longer term (12–24 months), offering the Healthy Heart programme to high-cardiovascular-risk patients did not improve their lifestyle behaviour nor cardiovascular risk and was not cost-effective on a population level.

## 1. Introduction

Cardiovascular diseases cause a high burden of morbidity and rising health care expenditures [1,2]. They are known to be mainly caused by unhealthy lifestyle behaviour such as poor-quality diet, insufficient physical activity and smoking. Therefore, the key component of prevention and proactive clinical treatment might comprise changing unhealthy lifestyle [3]. Thereby, potentially more than one cardiovascular risk factor can be improved at the origin.

Lifestyle changes as part of a cardiovascular risk management approach can be initiated at the level of primary care, for example by the general practitioner (GP), but may further be guided by other dedicated lifestyle professionals. This process could potentially be supported by health care professionals following the needs and possibilities of the patient. When available, GPs can refer to combined lifestyle intervention programmes, provided by lifestyle coaches who help patients to improve multiple lifestyle factors. Structured examples of such programmes are suggested to reduce cardiometabolic risk [4,5,6].

Several combined lifestyle interventions in The Netherlands have been recognised as being effective for a 1–3 kg weight reduction in people with obesity in the short term (1 year) and longer term (1.5–2 years) [7,8,9]. However, other cardiovascular risk factors such as blood pressure were only examined to a limited extent. Furthermore, implementation still faces several barriers [9,10,11,12] and may be more difficult in a socio-economically diverse urban area [13,14]. Additionally, the evidence of the cost-effectiveness of lifestyle programmes is inconclusive, especially when analysed on a population level [15,16,17].

Initiation of a healthy lifestyle is often considered to be time-consuming and influenced by the motivation and commitment of both patients and care providers [18,19,20]. In previous studies intention- and goalsetting appeared to have a positive effect on behaviour change [20,21] and should, therefore, be the starting point of all lifestyle changes. Awareness of the importance of lifestyle in the treatment and secondary prevention of cardiovascular disease, by both the care provider and patient, may act as a window of opportunity [22] leading to lifestyle change [23]. Discussing the importance of a healthy lifestyle, and the possibility to be supported by a combined lifestyle intervention during a consultation, may encourage behavioural change and thus lead to positive changes in lifestyle behaviour, even if a patient is not referred to a lifestyle programme. However, the effect and cost-effectiveness of implementing such a lifestyle programme on practice level in primary care is unknown.

Therefore, in this study we aimed to examine the (cost-)effectiveness on a population level of offering referral to a combined lifestyle programme—“Healthy Heart”—as part of cardiovascular risk management in primary care in an urbanised, socio-economically diverse area. We examined the effect of offering the programme on weight, systolic blood pressure (SBP), cholesterol levels, diet quality, physical activity, smoking and alcohol consumption in the target population. Furthermore, we examined goalsetting on lifestyle behaviour and self-reported achievement of goals. To examine the cost–benefit of implementing the programme we performed a cost–utility analysis. With this study, we provide new insights on the effects of offering a lifestyle intervention in a real-life primary care setting.

## 2. Materials and Methods

### 2.1. Study Design and Setting

A detailed description of the study and Healthy Heart programme was published elsewhere [24]. In short, this study is a practice-based, non-randomised stepped-wedge cluster trial including 56 primary care practices (clusters) in the region of The Hague (the Netherlands) with ~1,000,000 inhabitants. During the intervention period, potential participation in a combined lifestyle programme called “Healthy Heart” was discussed with participants and referral to the programme was offered as part of a structured cardiovascular risk management programme. Participants could either be referred or choose to receive regular care without support by the lifestyle coaches. During the control period, participants received regular care. The active trial recruitment period was between June 2017 and April 2019: each practice could recruit patients for study participation for one year in total with first a 2–6 month control (regular care) period and then a 4–10 month intervention (optional referral) period (Figure 1).

### 2.2. Participants

Adults (>18 years) eligible to participate in the study were estimated to be at high risk for the development of cardiovascular disease (>10% 10-year cardiovascular morbidity risk) [25], and were recruited during routine cardiovascular preventive care by their GP or practice nurse (PN). Persons with pre-existing cardiovascular disease, diabetes mellitus, or significant or palliative comorbidities were not recruited. After oral and written informed consent, participants filled in questionnaires. Patients recruited during the intervention period discussed their lifestyle goals with their PN or GP and proceeded, based on shared decision-making, with standard care only or were *in addition* referred to a certified lifestyle coach who offered the Healthy Heart programme. During the control period, participants were provided standard preventive care [25]. Questionnaire data were linked to routine primary electronic health record (EHR) data for the analysis. During the full recruitment period, participants were also given the option to provide informed consent for use of their EHR data only, without filling in questionnaires, while receiving standard care.

#### 2.2.1. Healthy Heart Lifestyle Programme

The five-month lifestyle programme to which participants could be optionally referred was predesigned by the care group based on available evidence. It consisted of eight group sessions (8–10 persons) and three individual sessions supervised by a lifestyle coach, in which all aspects of lifestyle behaviour change (dietary quality, overweight, physical activity, smoking, alcohol intake and stress management), motivation and personal goals on behaviour change were addressed. The programme took place in patients’ own neighbourhoods and was free of charge.

#### 2.2.2. Outcomes and Sources Effectiveness Study

The primary outcomes were lifestyle behaviour—and cardiovascular risk factors (described below) in the period 3–6 months (short term) and 12–24 (longer term) months after baseline. The secondary outcome was the number of cardiovascular-risk-management (CVRM)-related primary practice consultations in the past 6 months and the difference in weight, physical activity, dietary quality and alcohol usage between those who set a goal at baseline for these outcomes and those who did not. Outcomes were measured at baseline, 3, 6, 12 and 24 months from digital or hard-copy questionnaires or continuously between baseline and 27 months from the EHR. An overview of all measurements is displayed in Appendix A.

Outcomes derived from questionnaires were total minutes of physical activity per week (Short Questionnaire to Assess Health-enhancing physical activity), dietary quality (total score from the Dutch Healthy Diet (DHD) index; higher scores indicate higher diet quality) and smoking (never/previous/current smoker). Patient-reported smoking cessation was defined as being a smoker at baseline and reported ‘no smoking during the last 7 days’ (3–6 months) or ‘no smoking during the last 6 months’ (12–24 months) at follow-up measurements [24].

Outcomes derived from the EHR were systolic blood pressure (SBP) (mmHg), LDL-cholesterol (mmol/L), HDL-cholesterol (mmol/L). EHR measurements were selected as the nearest to the exact date of baseline, baseline plus 3, 6, 12 and 24 months in a six-week range around baseline, baseline plus 3 and 6 months and a two-month range around baseline plus 1 and 24 months. CVRM-related visits during the 6 months prior to baseline, baseline–6 months, 6–12 months, 12–18 months and 18–24 months, respectively, were derived from the EHR, reasoning that a record of an anamnestic (e.g., alcohol intake), SBP measurement or laboratory test indicated either a live or telephone visit to the primary practice.

Weight and alcohol consumption were primarily derived from questionnaires (patient-reported weight (kg) and mean number of units of alcohol per day, calculated from the DHD-index). When patient-reported data were missing, the data were filled in with weight (kg) registrations in the EHR registered with ‘average alcohol use’ in the EHR.

For the EHR-derived data, all registered measurements between 3–6 months and 12–24 months were included in the effect analysis.

#### 2.2.3. Utilities, Healthcare Use and Costs

Utilities reflect the valuation of quality of life, on a scale from zero (as bad as death) to one (perfect health). Utility scores at baseline, 6, 12 and 24 months were calculated using the 5-level EQ-5D (EQ-5D-5L) using the Dutch tariff [26]. As sensitivity analyses, utilities were calculated from the SF-6D, as calculated from the SF-12 questionnaire (US version) [27] and the EQ-5D visual analogue scale (VAS), using the transformation 1 − (1 − VAS/100)^1.61^ [28].

Health care use in the preceding 6 months (including GP, medical specialists, dietician, physiotherapist, lifestyle coach (individual or in group), home and hospitalisation care) was patient-reported using questionnaires at baseline and, respectively, 6, 12, 24 months after baseline. Healthcare use between 12 and 18 months was interpolated from the questionnaires at 12 and 24 months. Cardiovascular medication prescriptions and laboratory procedures during the period baseline–24 months were derived from the EHR.

Costs from a healthcare perspective were calculated in Euros at price-level 2021. Intervention costs included the individual participant costs for the Healthy Heart programme (EUR 434.59 per participant) and an information meeting for primary practices (189.59 per participant). Other health care was valued using Dutch standard prices for medication [29] (assuming generic products with standard daily dosage, plus administrative costs per prescription), for laboratory procedures [30] and for other health care [31].

#### 2.2.4. Determinants

Baseline measurements were age, sex (patient-reported or from EHR) and origin (Dutch/non-Dutch), educational status (low (no education or vocational training)/high (college or university)), living status (living alone/co-habiting), job status (currently employed/non-employed (<65 years)/retired) (patient-reported), comorbidities (EHR). Neighbourhood liveability index, as measure of socio-economic position, was derived from governmental publicly available data [32]. Comorbidities (yes/no) were defined as active diagnosis registered with International Classification of Primary Care (ICPC) codes before 3 months after baseline and were reported based on their prevalence in the study population and clinical relevance. Comorbidity groups were: hypercholesterolaemia, hypertension, vascular, psychiatric and ‘other chronic’ comorbidities [33]. Self-efficacy was measured by the General Self-Efficacy Score and quality of life by the EQ-5D-visual analogue scale (VAS) (Appendix A).

Goalsetting (yes/no), motivation (0–10 scale) and self-confidence of achieving this goal (0–10 scale) for each lifestyle goal (more exercise, improve diet quality, intentional weight loss, lower alcohol intake, quit smoking) were assessed at baseline. Self-perceived achievement (yes/no) of these goals was assessed at 3 and 6 months.

### 2.3. Statistical Analyses

#### 2.3.1. Effectiveness Study

Participants recruited during two observation periods were compared: the control period (all patients receiving standard care) versus the intervention period (all patients receiving standard care and who were offered to participate in the Healthy Heart programme). A third group of patients, who did not complete questionnaires, served as an extra control group for the outcome measurements derived from the EHR, as this group was not biased by motivation for completing or the content of the questionnaires. The baseline measurement for this group was set as their first GP visit after 1 July 2017 (start of the recruitment period).

Differences between the intervention and control period in goalsetting and self-perceived achieved goals were assessed with chi-square tests and differences in motivation and confidence in lifestyle changes were assessed with independent *t*-tests.

To examine the outcomes, multivariate linear, ordinal (alcohol) and Poisson (number of visits to primary practice) mixed models were used. Mixed models for (1) the period 3–6 months and (2) 12–24 months after baseline were calculated with period (control/intervention) as fixed effect and with random effects for participant and practice, adjusted for baseline measurement of the outcome, sex, age, antihypertensive- (SBP model) and lipid-lowering medication usage (LDL- and HDL-cholesterol model) and total general self-efficacy scale (alcohol model). For weight, SBP and cholesterol measurements, a period of 1.5 months–7.5 months and 10–26 months was included in the analysis to include all in-between measurements in the models. For CVRM-related practice-consultations, the timeframes 0–12 and 12–24 months were analysed. To assess effect modification with sex and age, an interaction term with group and these determinants were added to the models. Due to a limited number of current smokers, difference between the control and intervention period on smoking cessation (yes) was examined using logistic regression analysis with 7-day abstinence at 3 or 6 months and 6-month abstinence at 12 or 24 months.

We performed two in-depth analyses. First, in order to examine the effect of the Healthy Heart programme itself, we repeated the models with participants who participated in the programme compared to those who did not (recruited both during the control and intervention period). Additionally, to assess external validity, we repeated the models regarding weight, SBP and cholesterol-levels of participants recruited during the intervention period compared to participants in the EHR-group for 12–24 months. Since routine consultations occur annually, measurements at 3–6 months in the EHR are likely to mainly include suboptimal outcomes.

#### 2.3.2. Economic Evaluation

Difference in costs and QALYs between the intervention and control period were estimated using linear regression analysis, adjusted for sex, age, recruitment period (control/intervention) and baseline values of costs and utilities. Cost-effectiveness acceptability curves (CEACs) were plotted, showing the probability that the intervention is cost-effective compared with the control period, for a range of threshold values for willingness to pay (WTP) per additional QALY [34].

#### 2.3.3. Missing Data

For the economic evaluation, missing data were imputed using multiple imputation, with 100 imputed datasets. Predictors in the imputation procedure included inclusion period (control/intervention), sex, age, EQ-5D utilities, medication costs, GP cardiovascular risk management (CVRM) costs, GP other costs and total medical specialist outpatient costs (Appendix A).

Analyses were performed with SPSS version 25.0.02 and R version 2022.02.0, lme4 and ordinal packages.

## 3. Results

### 3.1. Recruitment and Characteristics

A total of 511 patients were included during the control period, and 276 during the intervention period, of which 451 (88%) and 251 (91%) could be linked to EHR data. During the intervention period, 40 persons (15%) participated in the Healthy Heart programme. A total of 155 participants were included in the EHR-only group. Persons who did not complete the questionnaires, nor could be linked to the EHR, were excluded from the analysis (*n* = 16 and for the economic evaluation *n* = 17) (Figure 2).

Baseline characteristics were comparable between the periods (Table 1). Overall, over half of the population (mean age 65.0 ± 9.6) were women (*n* = 442 (56%)), most were of Dutch origin (*n* = 659 (88%) and lived in a neighbourhood with high liveability index (*n* = 497 (72%)). There were no differences between the groups in numbers of missing data, except for the questionnaire drop-out rate at 24 months (intervention: 93 (34%) vs. control: 123 (24%)). Missing data in the patient-reported outcomes increased during follow-up. In the EHR-derived outcomes, missing data were highest at 3 and 6 months (Appendix A).

### 3.2. Cardiovascular and Lifestyle Behaviour Outcomes

Trends in outcomes between baseline and 24 months are visualised in Figure 3 and estimates of effects are presented in Table 2. Weight, blood pressure, LDL-cholesterol and HDL-cholesterol, physical activity and alcohol usage did not show a clear change in trend over time and did not differ between the participants recruited during the control and intervention period during 3–6 months nor 12–24 months. Dietary quality increased over time in both periods, but with no differences between the periods (β (95% CI): 3–6 months: 0.95 (−0.93; 2.83), 12–24 months: 1.54 (−0.57; 3.64)). Participants in the intervention period were as likely as participants in the control period to quit smoking (OR (95% CI) 3–6 months: 2.54 (0.45; 14.24) 12–24 months: 1.07 (0.10; 11.64)). Number of CVRM-related consultations did not differ between the periods during 0–12 and 12–24 months (OR (95% CI): 0–12 months: 1.00 (0.91; 1.11) 12–24 months: 0.99 (0.83; 1.18)).

### 3.3. Effect Modification of Sex and Age

Stratified analyses showed less minutes of physical activity for the intervention period compared to the control period in those aged <65 years (mean difference β (95% CI) −276 (−536; −18) and more minutes of physical activity for the intervention period for those aged ≥65 (mean difference β (95% CI) 231 (84; 377) (interaction term: β (95% CI) 21 (7.1; 36) *p* = 0.004). This difference disappeared during 12–24 months. Regarding CVRM-related consultations during 3–6 months, women in the intervention period had less consultations than those in the control period (OR (95% CI): 0.87 (0.77; 0.99). In men, there was no difference between the periods (OR (95% CI): 1.11 (0.96; 1.28)) (interaction term: OR (95% CI): 0.79 (0.65; 0.95) *p* = 0.011). No other effect-modifying effects were found.

### 3.4. Goalsetting and Outcomes

Most of the participants (*n* = 575 (77%) set one or more goals in the questionnaire at baseline. The percentages of set goals ranged between 69% (more exercise) and 26% (alcohol reduction) (Table 3). The number of participants with a goal for weight reduction was higher (69%) in the intervention group compared with the control group (57%) (Pearson chi-square: 10.1, *p* = 0.001). Motivation for weight loss and diet was higher in the intervention group, compared to the control group, but not for physical activity, alcohol reduction and smoking cessation (Table 3). Confidence in achieving the goal was higher for all lifestyle goals in the intervention group, except for smoking cessation.

Percentage of patient-reported achieved goal at 6 months was highest for alcohol reduction (77–84%) and lowest for weight reduction, with a significant difference between the control and intervention group (57 out of 204 (28%) vs. 42 out of 93 (45%) (Pearson chi-square: 8.17 *p* = 0.017)).

For weight, physical activity, alcohol usage and smoking cessation, cardiovascular and lifestyle outcomes of those who set a goal at baseline were similar at 3–6 months and 12–24 months to those who did not set a goal. Those who set a goal ‘to improve dietary quality’ had a significantly higher dietary index score at 3–6 months compared to those without a goal (β (95% CI): 3.1 (1.2; 4.9), but this difference disappeared during longer-term follow-up (12–24 months: β (95% CI): 2.0 (−0.1; 4.0)) (Appendix A; Figure 1).

### 3.5. Economic Evaluation

Participants’ utilities at baseline were comparable between the control and intervention period (adjusted mean difference (95% CI) −0.004 (−0.04; 0.03)) (Table 1). First-year QALYs were significantly lower in the participants in the intervention group (mean difference (95% CI) −0.06 (−0.11; −0.02)), while in the second year, the difference was non-significant (Table 2, Appendix A). Also for the total study period, QALYs were lower in participants during the intervention period, but the association did not reach statistical significance (intervention: 1.38 ± 0.88, control: 1.49 ± 0.79, adjusted difference (95% CI) −0.10 (−0.20; 0.002)). QALYs based on the SF-6D questionnaire and EQ-VAS showed similar results (Figure 3 and Appendix A).

### 3.6. Costs

Baseline costs (6 months prior to study period) were comparable between the periods (adjusted mean difference (95% CI) −18 (−247; 212) *p* = 0.881) (Table 1). With 15% of the patients participating in the Healthy heart programme, average intervention costs were estimated at EUR 91 per patient in the intervention period (95% CI 70–108). Mean costs of 2-year total CVRM care were comparable (mean ± SD intervention: mean adjusted difference (95% CI) EUR 106 (−80; 293) *p* = 0.266)). Mean 2-year total health care costs were mean ± SD EUR 2909 ± 4826 (intervention) and EUR 2457 ± 3494 (control), with no significant difference between the periods (mean adjusted difference (95% CI) EUR 484 (−110; 1079) *p* = 0.110) (Appendix A).

Combining the difference in QALYs and costs, the probability that offering the Healthy Heart programme in primary practices is cost-effective compared to not offering the programme was below 6%, regardless of the willingness-to-pay per QALY (Appendix A). In the sensitivity analyses using the SF-6D and EQ-VAS to calculate QALYs, this probability remained below 6% (SF-6D) and 26% (EQ-VAS).

### 3.7. In-Depth Analysis

#### 3.7.1. Participants of the Healthy Heart Programme

Participants of the Healthy Heart programme (*n* = 40), compared to those who did not participate (*n* = 747), showed a higher DHD index at 3–6 months and 12–24 months (group difference (95% CI): 8.74 (4.6; 12.9) and 7.29 (2.2; 12.2)), a slightly lower HDL-cholesterol (group difference (95% CI): −0.09 (−0.17–0.01)) during 3-6 months and used less alcohol (OR 95% CI 0.66 (0.065–0.66)) during 3–6 months but not significantly during 12–24 months (OR 95% CI 0.89 (0.08–9.66)) (Appendix A).

#### 3.7.2. Participants with Only EHR Data

No differences between participants in the intervention period and participants who did not fill in the questionnaires (EHR-only group) were found regarding weight, SBP and cholesterol levels (Appendix A).

## 4. Discussion

In this study, we evaluated the (cost-)effectiveness on population level of offering referral to the lifestyle programme “Healthy Heart” in a high-risk cardiovascular disease population in primary care. The most important finding is that for both the shorter (3–6 months) and longer term (12–24 months), there were no differences in cardiovascular or lifestyle behavioural outcomes between the control period versus the intervention period on a population level. Furthermore, there was no effect on number of cardiovascular-related practice consultations and offering the Healthy Heart programme did not improve QALYs or reduce costs in the intervention period. Although a majority of the participants set a goal to change their lifestyle behaviour, only participants with a goal to improve their diet quality slightly improved their diet quality in the short term, but not in the long term.

### 4.1. Comparison with Previous Literature

We assumed that introducing a lifestyle programme in primary practices would give lifestyle change a boost during the intervention period and participants would be stimulated to change their lifestyle through goalsetting and discussion of participating in the programme. However, a significant short-term or longer-term (24 months) effect on cardiovascular outcomes or lifestyle behaviours was not found.

The most important difference between the current study and previous studies is that previous studies examined the actual participants of a lifestyle programme, while in our study, outcomes were measured on a population level, where participants had the possibility to participate in the programme—or not. It appeared that only 15 percent of the participants included in the intervention period were willing to participate in the Healthy Heart programme. In line with previous studies [6,7,8], the 2-year effect on the examined outcomes was limited among those who participated in the Healthy Heart programme during our study, although the analysis may have lacked power due to the small number of participants in this subgroup. Furthermore, the five-month supported programme may have been too short to result in significant changes. However, currently implemented Dutch lifestyle interventions with an intensive treatment period of 6–8 months have been proven effective in BMI reduction [6,8]. Total follow-up time was 2 years: if participants of the programme would have maintained their lifestyle adaptations, one should expect improved outcomes for those participated in the programme.

In contrast, the comparable Spanish EIRA trial [35] implemented a lifestyle intervention on an individual, group-based and community level. Although it was, like our study, not cost-effective [17], it showed positive outcomes on lifestyle behaviour in the intervention group on a population level. The personalised approach based on stage of motivation may have contributed to the positive effects of the EIRA trial, where in our study, one group-based programme was offered for all.

Van der Bruggen et al. [15] suggested in a large modelling study that offering lifestyle interventions to people at risk could be cost-effective, in contrast to offering the intervention to the full population. However, our study showed—in line with the EIRA trial [17]—similar (CVRM-related) costs and similar or worse QALYs in the intervention period compared to the control period. The lower utilities in our intervention period may be partly explained by the stepped-wedge design, which may have induced selection bias. The low percentage of participants in the programme may have resulted in a lack of power in detecting a difference in between the intervention and control period in health care costs. Additionally, the effects of prevention of cardiovascular disease are expected in the long-term, so longer follow-up (>10 years) may reveal different results.

Because intention supports behaviour change, participants in our study were challenged—either through questionnaires (control period) or by their GP/PN (intervention period)—to set a goal on lifestyle change at baseline [20]. However, the number of achieved goals at 3 and 6 months was limited, which is a common pattern in behaviour changes [36]. People who did not complete the questionnaires (EHR-only group), had similar cardiovascular outcomes as people in the intervention period, indicating that goalsetting at baseline alone is not enough for cardiovascular risk reduction through behaviour change. Possibly, people did not receive the right support to achieve these goals, even when there was a focus on lifestyle change within the practices during the intervention period. Moreover, despite motivation, the barriers were probably too high to participate in the Healthy Heart programme [13], which could have helped achieve their goals [37].

This study confirms that implementation of lifestyle programmes in primary care still faces large barriers [11,38,39]. Our results are in line with in the recently introduced combined lifestyle intervention for people with obesity [12], which has been covered by basic health insurance since 2019, where the 1-year effects on BMI were limited. Furthermore, number of dropouts of this programme was 26% and the number of participants in our region, South Holland, are relatively low. Since our study is fully based in “running” practice, it may better reflect the effect that will be realised in a real-life primary care setting than previous trials could [6,8,40]. The (cost-)effectiveness assessment of our study on a population level can support GPs and policymakers in deciding about implementation of such programmes. On the other hand, the implementation only lasted during the study period, and it is known that it takes time to implement new health policies well [41]; thus, it may have been too short to see results.

Time, effort and lack of knowledge about the effectiveness of the programme were barriers for care providers to promote participation in the Healthy Heart programme [42] indicating that although lifestyle programmes are available within the region, both patients and care providers should be willing to participate for successful implementation and recruitment [43]. Higher volumes of usage of physiotherapy and dietician care may indicate a shift towards an improved focus on lifestyle change; however, the difference was not significant (data in Appendix A). Adaptation by care providers may be influenced by lack of knowledge about the effect of the programme, scientific evidence, availability of lifestyle coaches or the belief that healthcare authorities may be better equipped to provide preventive care [39]. Furthermore, people who were not motivated to participate in the Healthy Heart programme may demand a more individualised approach to address lifestyle behaviour change [12,43], such as in the trial of Zabaleta-Del-Olmo [35]. A longer implementation period in our study could possibly have supported primary care providers to gain more confidence with the programme.

### 4.2. Limitations

Missing data are a common problem in longitudinal and observational studies, especially when they are fully embedded in routine clinical practice. As both questionnaires and EHR-data were used, two different missing data patterns could be observed: first, through dropouts of the questionnaires, and second, to registration and attendance bias in the EHR. Missingness was addressed by the use of mixed models for the effect analysis and multiple imputation for the cost-effectiveness analysis, both common methods for similar studies. Concerning potential registration bias in the EHR data, we controlled for practice-variance in our models. Bias through non-attendance might have under- or overestimated the effects. Non attendees could be care avoiders with poor outcomes, and on the other hand be participants with good disease control and, therefore, not intensively monitored. However, registration patterns were similar for the control and intervention period; thus, bias on the effect estimates between the groups will be limited.

We did not recruit a fully representative sample of the high-risk population in our region. Results may, therefore, not be generalisable to other regions. Participating practices and the lifestyle programme were located throughout socio-economically diverse neighbourhoods in an urbanized region, while, compared to the target population, the number of people included in the study with a migrant background or lower education was relatively low [44]. Regarding weight reduction, the mean weight of our population was representative for the target population [45]. However, generally overweight people with type 2 diabetes [3] were not recruited, in whom the largest effect of a weight-reduction intervention may have been found. Furthermore, the percentage of smokers in our study was low; therefore, the effect of the trial on quitting smoking could only be estimated to a limited extent. This low percentage is in line with the percentage of smokers in The Hague aged > 65 years or with a higher educational level, thus is representative for the population that was reached by this study [44,46]. However, higher percentages of smokers are known to be among people within groups with a migration background and with lower educational levels, who were not fully reached. Lastly, the number of participants already meeting the Dutch guidelines for healthy alcohol consumption (1 or less glasses of alcohol per day) is slightly higher than in the general population of The Hague (The Hague: 51% [47], control group: 61% and intervention group 68%), thus not meeting the target high-risk population, namely those with high alcohol consumption.

### 4.3. Future Perspectives

Organisation and incorporation of an integrated lifestyle programme in primary practice is not sufficient yet to help patients with a high cardiovascular risk towards a longstanding healthy lifestyle. First, not all people eligible for combined lifestyle programmes are reached: future research should focus on reaching a larger (socially diverse) population, how to motivate them and how to refer effectively to a local lifestyle intervention initiative. Furthermore, in primary practice, only persons with an indication for indicated or health-related prevention are reached. The role of the social and governmental domain in universal prevention of cardiovascular risk should be examined instead of focusing on the healthcare domain only. For indicated prevention in primary care, we suggest a close collaboration with lifestyle coaches and other relevant providers, while GPs and practice nurses may play a key role in identifying patients at risk and susceptible for intervention.

## 5. Conclusions

In conclusion, the offering of the Healthy Heart programme to patients with high cardiovascular risk in The Hague in the current form did not lead to improvement of cardiovascular risk outcomes, lifestyle behaviour or number of practice-consultations and it was not cost-effective. Further research should focus on how to reach the whole at-risk population, also beyond primary practice, and how to implement lifestyle change programmes as a proactive intervention in primary care in collaboration with other domains and thereby reduce the cardiovascular disease burden.

## Figures and Tables

**Figure 1 ijerph-20-05040-f001:**
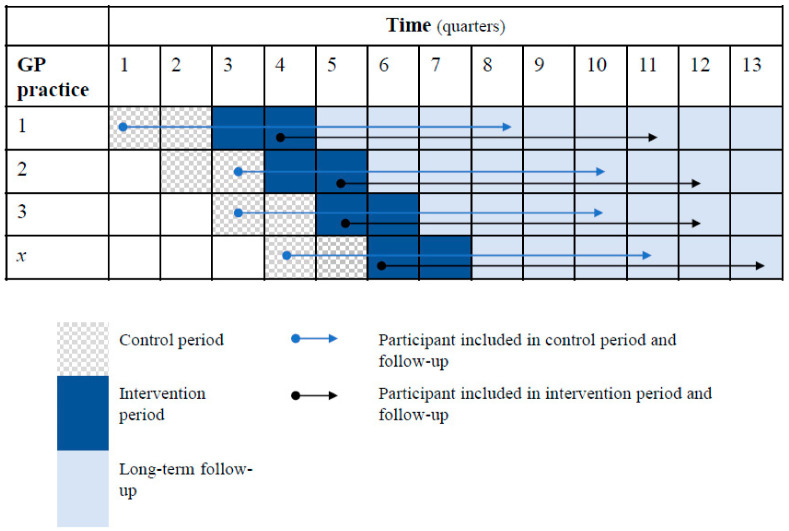
Stepped-wedge design, inclusion and follow-up of participants during the control and intervention period.

**Figure 2 ijerph-20-05040-f002:**
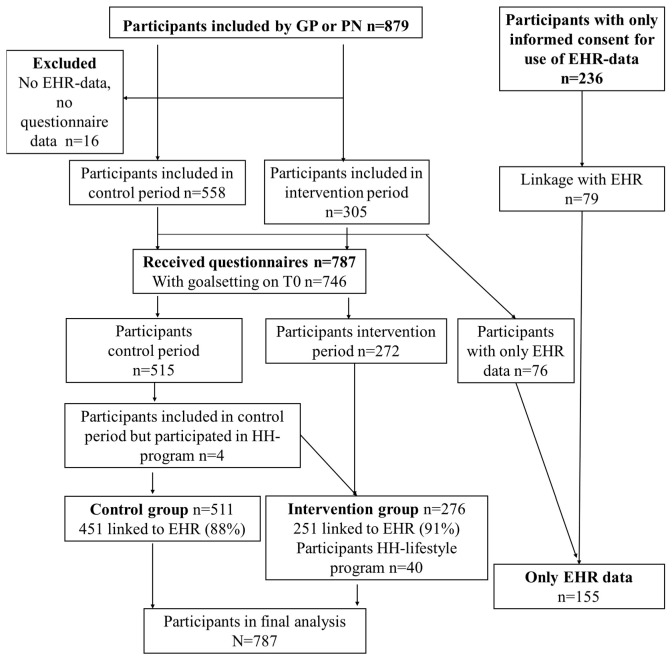
Flow-chart patient inclusion. Legend: PN: practice nurse, GP: general practitioner, EHR: electronic health record, HH-lifestyle programme: Healthy Heart lifestyle programme.

**Figure 3 ijerph-20-05040-f003:**
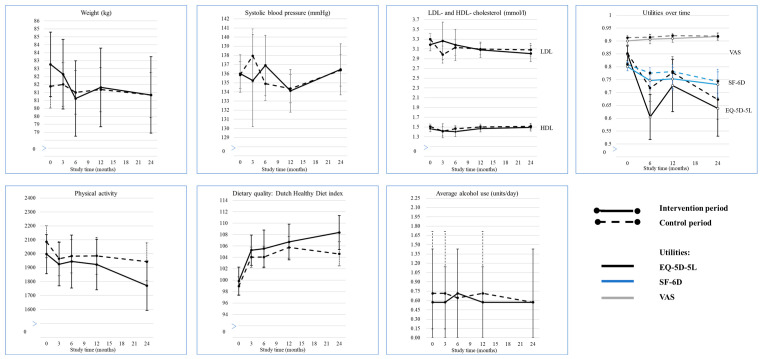
Twenty-four-month trend in cardiovascular risk factor and lifestyle behaviour outcomes of the Healthy Heart intervention. Legend: Outcomes are displayed as mean (95% CI) and alcohol usage as median (IQR) per time measurement.

**Table 1 ijerph-20-05040-t001:** Baseline characteristics of included participants of the Healthy Heart trial.

Characteristic	Available Data (*n*)	Period (N = 787)			
		Intervention (*n* = 276)		Control (*n* = 511)	
		*n*, Unless Otherwise Indicated	%, SD or IQR	*n*, Unless Otherwise Indicated	% or SD
**Age** (years)	784	64.5	10.0	65.6	9.3
**Sex** (women)	785	159	58%	283	56%
**Origin** (Dutch)	743	225	86%	434	90%
**Educational level** (low)	724	151	59%	283	61%
**Household composition** (cohabiting)	744	194	74%	355	74%
**Job status**	744				
Currently employed (<65 years)		90	35%	172	36%
not employed (<65 years)		41	16%	58	12%
retirement (>65 years)		130	50%	253	52%
**Neighbourhood deprivation score**	691				
Weak		27	11%	57	13%
Satisfactory		30	13%	80	18%
Good–extremely good		183	76%	314	70%
**Comorbidity ***	690				
Chronic comorbidity		22	9.2%	34	7.5%
Hypercholesterolaemia		88	37%	145	32%
Hypertension		196	82%	392	87%
Macrovascular disease		11	4.6%	15	3.3%
Impaired kidney function		30	13%	60	13%
**Medication usage (yes)**	691				
Antihypertensive		177	74%	351	78%
Lipid lowering		93	39%	200	44%
Anti coagulants or thrombocyte aggregation inhibitors		15	6.3%	45	10%
Psychiatric medication		18	7.5%	41	9.1%
**Body mass index (kg/m^2^)**	778	28.0	4.8	27.5	4.5
**Use of alcohol** (yes)	772	199	74%	398	79%
**Smoking status**	742				
Never		120	48%	225	47%
Previous		107	43%	213	45%
Current		23	9.2%	39	8.2%
**General self-efficacy scale (scale 10–40)**	733	33	(30–37)	33	(30–37)
**Quality of life (scale 0–100)** Perceived health, EQ-VAS score	732	78.8	14.1	80.9	14.6
**Adherence to exercise guideline ** (yes)**	730				
minutes of physical activity		97	38%	181	38%
minutes of physical activity + strength and balance		87	34%	157	33%
**Utility *****					
EQ-5D-5L	786	0.85	0.28	0.85	0.32
SF-6D	786	0.80	0.13	0.81	0.14
VAS	786	0.90	0.12	0.91	0.11
**Total health care costs (Euros) *****	786	EUR 651	1423	EUR 671	1567

EQ-5D-5L: EuroQol—5 Dimensions—5 Levels, SF-6D: Short Form—6 Dimensions, VAS: Visual Analogue Scale. Continuous values are presented as means ± standard deviation or median (IQR) (interquartile range). Categorical values are presented as *n* (% of participants with non-missing data within group). * Comorbidities were: hypercholesterolaemia (T93), hypertension (K85, K86, K87); vascular: nephropathy (U99.01), kidney failure according to laboratory measurements (mild–high increased risk on CVD risk/progression/mortality41), retinopathy (F83), neuropathy (N94.02), angina pectoris (K74), myocardial infarction (K75), other ischemic diseases (K76), transient ischemic attack (TIA) (K89), cerebral infarction (K90), peripheral artery disease (K92.01), heart failure (K77); psychiatric: alcohol or drug addiction (P15, P19), dementia (P70), anxiety and psychosis (P72, P74, P79), mood disorders (P03, P73, P76), stress (P01, P02), other (P77, P78, P80, P99, P06, P85), social problems (all ICPC codes within ‘Z’ group) and other chronic comorbidities, based on its high prevalence in the Dutch population [32]. ** Dutch physical activity guidelines for adults (Dutch Health Council, 2017): ≥150 moderate-intense activity, and 2 times per week strength and balance exercises. *** results of 100 pooled imputed datasets.

**Table 2 ijerph-20-05040-t002:** Cardiovascular and lifestyle outcomes of participants of the Healthy Heart intervention with and without goalsetting.

	Control Period Versus Intervention Period						Goal Versus No Goal at Baseline					
Outcome	Control Period: Baseline		Intervention Period: Baseline		Between- Period Δ after Follow-Up: Adjusted β or Odds (95% CI) ^b^		Goal (No)		Goal (Yes)		Between- Period Δ after Follow-Up: Adjusted β or Odds (95% CI)	
	Mean, Median or *n* ^a^	SD, IQR or %	Mean, Median or *n* ^a^	SD, IQR or %	3–6 Months	12–24 Months	Mean, Median or *n* ^e^	SD, IQR or %	Mean, Median or *n* ^e^	SD, IQR, or %	3–6 Months	12–24 Months
Weight (kg)	81.4	15.8	82.8	17.0	−0.5 (−1.1; 0.05)	−0.13 (−1.0; 0.7)	73.5	12.4	87.1	16.1	−0.41 (−2.2; 0.2)	0.75 (−1.7; 0.2)
Systolic blood pressure (mmHg)	136	15	136	15	0.15 (−2.7; 3.0)	−0.35 (−2.3; 1.6)	n/a		n/a		n/a	n/a
LDL-cholesterol (mmol/L)	3.3	1.0	3.2	0.9	0.07 (−0.2; 0.35)	0.02 (-0.1; 0.2)	n/a		n/a		n/a	n/a
HDL-cholesterol	1.5	0.4	1.5	0.4	−0.03 (−0.10; 0.05)	−0.01 (−0.05; 0.3)	n/a		n/a		n/a	n/a
Minutes of total weekly physical activity	2085	1297	1997	1138	38 (−97; 171)	−42 (−201; 116)	1997	1168	2089	1270	58 (−75; 190)	46 (−104; 195)
Dietary quality (DHD index, scale 0–150)	99	18	100	20	0.92 (−1.0; 2.8)	1.47 (−0.6; 3.6)	100	18	99	18	**3.1 (1.2; 4.9)**	1.9 (−0.2; 3.9)
Alcohol intake ^c^					0.81 (0.4; 1.5)	0.66 (0.3; 1.7)					0.84 (0.4; 1.9)	2.17 (0.7; 7.1)
No alcohol	111	22%	74	27%			168	23%	436	59%		
0–1	201	39%	112	41%			50	31%	242	58%		
1–2	105	21%	47	17%			51	32%	95	23%		
2 or more	83	16%	36	13%			54	34%	59	14%		
Quit smoking during follow-up ^d^	43	8.4%	19	6.9%	2.54 (0.5; 14.2)	1.07 (0.1; 11.6)	32	52%	30	48%	0.42 (0.1; 2.5)	0.30 (0.03; 3.2)
Number of CVRM-related contacts with practice in the past 6 months	1	1–2	2	1–2	0.99 (0.9; 1.1)	0.92 (0.8; 1.1)	n/a		n/a		n/a	n/a

DHD-index: Dutch Healthy Diet index, CVRM: cardiovascular risk management, HDL: high-density lipoprotein, IQR: interquartile range, LDL; low-density lipoprotein, n/a; not applicable, SD: standard deviation. Participants per group included at baseline: control period (*n* = 511); intervention period (*n* = 276). **^a^** Percentages are of all participants per period. ^b^ Reference group: control period. Models are adjusted for: sex, age, baseline value of outcome (all), antihypertensive medication (blood pressure model), lipid-lowering medication (LDL and HDL models), total score of General Self-Efficacy Scale (alcohol). For alcohol intake, quit smoking and number of CVRM visits, between period differences are displayed in odds ratio (95% CI). ^c^ Average units per day; reference group: no alcohol. ^d^ Numbers displayed at baseline are participants who indicated to be a current smoker at baseline. ^e^ For smoking: numbers of those with, and without a goal at baseline of total current smokers (*n* = 62). Δ: difference. Bold numbers indicate a *p*-value < 0.05. n/a = Not applicable.

**Table 3 ijerph-20-05040-t003:** Goalsetting at baseline, patient-reported achievement of goals of participants of the Healthy Heart trial.

	Goal at Baseline (Yes) (N = 787)						Goal Achieved (Yes) after 6 Months ^a^				
	Control Period		Intervention Period		Missing ^c^	Mean Δ (95% CI)	Control Period		Intervention Period		Missing ^c^
Goal at Baseline (yes)	N, Mean, Median	%, SD, IQR	N, Mean, Median	%, SD, IQR	%		N, Mean, Median	%, SD, IQR	N, Mean, Median	%, SD, IQR	%
**Minutes physical activity per week**	297	62%	172	66%	5.8%		127	61%	65	68%	25%
Motivation	7.8	1.3	8.03	1.1		0.23 (−0.01; 0.5)					
Confidence	**7.2**	**1.5**	**7.55**	**1.4**		**0.32 (0.05; 0.6)**					
**Diet quality**	294	62%	162	62%	6.0%		147	75%	69	73%	26%
Motivation	**7.5**	**1.5**	**7.81**	**1.3**		**0.30 (0.03; 0.6)**					
Confidence	**7.2**	**1.5**	**7.55**	**1.4**		**0.37 (0.1; 0.7)**					
**Weight reduction**	**274**	**57%**	**181**	**69%**	5.8%		**57**	**28%**	**42**	**45%**	26%
Motivation	**7.8**	**1.5**	**8.12**	**1.4**		**0.29 (0.02; 0.6)**					
Confidence	**7.1**	**1.5**	**7.38**	**1.6**		**0.31 (0.02; 0.6)**					
**Alcohol usage ^b^**	101	(26%)	59	(30%)	6.1%		83	83%	34	77%	28%
Motivation	7.1	1.7	7.4	1.6		0.26 (−0.3; 0.8)					
Confidence	**7.3**	**1.7**	**7.9**	**1.3**		**0.53 (0.2; 1.0)**					
**Quit smoking ^b^**	24	55%	6	30%	5.7%		4	24%	1	50%	69%
Motivation	8.0	7.3–9.0	9	8.5–10		1.17 (−0.8; 3.1)					
Confidence	6.0	5.0–7.0	7.5	6.8–9.3		2.17 (−0.04; 4.4)					

IQR: interquartile range. Results are presented as number of measurements (% of total with goal) and motivation and confidence as mean ± SD or Median (IQR) of those who indicated to have a goal at baseline. ^a^
*n* (%) of those who set a goal at baseline. ^b^ Participants who indicated current alcohol use (*n* = 572) or current smoking (*n* = 62), respectively. ^c^ Participants who did not complete the questionnaire on this item. Bold numbers indicates a difference (*p* < 0.05) between the control and intervention group. Motivation scale 0–10. Confidence scale 0–10. Δ: difference.

## Data Availability

The datasets generated and analysed during the current study are available from the corresponding author on reasonable request.

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
