# Peer review of "The Effect and Cost-Effectiveness of Offering a Combined Lifestyle Intervention for the Prevention of Cardiovascular Disease in Primary Care: Results of the Healthy Heart Stepped-Wedge Trial"

_ijerph, 2023, doi:10.3390/ijerph20065040_

Round 1
Reviewer 1 Report
In this manuscript, the authors indicate The effect and cost-effectiveness of offering a combined lifestyle intervention for the prevention of cardiovascular disease in primary care: results of the Healthy Heart stepped-wedge trial The manuscript can be accepted after addressing the below mentioned corrections.
1. In introduction section, After sentences, ‘…and are known to be mainly caused by unhealthy lifestyle behaviour like poor.’’ It should be given information.
2. In introduction section, After sentences, ‘Therefore, the key component of prevention and proactive clinical treatment might comprise changing unhealthy lifestyle.’’ It should be given information.
3. In introduction section, After sentences, ‘Thereby, potentially more than one cardiovascular risk factor can be improved at the origin It should be given information. For this purpose the authors can look at the following articles for introduction section: Comparative biochemistry and physiology part C: toxicology & pharmacology226, 108608 and Biotechnology and Applied Biochemistry 69(3), 2273-2283
4. There are several English language issues. It should be corrected.
Author Response
We appreciate the time and effort that the reviewer has dedicated to provide valuable feedback on our manuscript. Thank you for giving us the opportunity to submit a revised draft of the manuscript. Please see below a point-by-point response to the reviewers’ comments and concerns.
Reviewer 1
Comment 1 & 2
In introduction section, After sentences, ‘…and are known to be mainly caused by unhealthy lifestyle behaviour like poor.’’ It should be given information.
In introduction section, After sentences, ‘Therefore, the key component of prevention and proactive clinical treatment might comprise changing unhealthy lifestyle.’’ It should be given information.
Response 1 & 2
Thank you for this notification. The reference was indeed missing. We inserted the following reference to this statement:
World Health Organization, Global status report on noncommunicable diseases 2014, 2014, ISBN 978 92 4 156485 4
Line 44: […] and are known to be mainly caused by unhealthy lifestyle behaviour like poor-quality diet, insufficient physical activity and smoking. Therefore, the key com-ponent of prevention and proactive clinical treatment might comprise changing un-healthy lifestyle (WHO 2014).
Comment 3
In introduction section, After sentences, ‘Thereby, potentially more than one cardiovascular risk factor can be improved at the origin It should be given information. For this purpose the authors can look at the following articles for introduction section: Comparative biochemistry and physiology part C: toxicology & pharmacology226, 108608 and Biotechnology and Applied Biochemistry 69(3), 2273-2283
Response 3
In addition to the ESC guideline (Visseren et al, 2021) following this statement , we added the following reference:
Lim, S. S., et al. (2012). "A comparative risk assessment of burden of disease and injury attributable to 67 risk factors and risk factor clusters in 21 regions, 1990–2010: a systematic analysis for the Global Burden of Disease Study 2010." The Lancet 380(9859): 2224-2260.
- There are several English language issues. It should be corrected.
We carefully checked and corrected the text on English language quality throughout the manuscript.
Reviewer 2 Report
This manuscript describes a clinical trial in which 787 patients with or without cardiovascular risks were enrolled, and lifestyle intervention were given to 40 patients with cardiovascular risks. The authors found no significant benefits in cardiovascular disease from the lifestyle intervention after a follow-up of up to 24 months. This study focuses on the short/long-term benefits of healthy lifestyle intervention in patients from the developed area and offers abundant data , which brings some scientific significance. I listed several major concerns need to be addressed.
1. The healthy lifestyle intervention was limited to 5 months. Is this long enough for the protective effect to be produced? Within such a short intervention window, the compliance of the patients is important. Is this healthy lifestyle intervention a supervised process? Or just self-regulated by the patients?
2. In the end, only 40 patients with cardiovascular risk meaningfully accepted the healthy lifestyle intervention. This small sample size is very likely to limit the power of the study and prevents a possibly true effect from being observed. How do the authors handle this issue?
3. According to the BMI and alcohol use, the overall lifestyle of the enrolled patients was not that hazardous. It’s not surprising due to the health education in the developed area like Netherland. Extra lifestyle intervention may not exert further effects on a prominent level. Did the authors take into account the correct sampled population?
Author Response
Reviewer 2
We appreciate the time and effort that the reviewer has dedicated to provide valuable feedback on our manuscript. Please see below a point-by-point response to the reviewers’ comments and concerns.
This manuscript describes a clinical trial in which 787 patients with or without cardiovascular risks were enrolled, and lifestyle intervention were given to 40 patients with cardiovascular risks. The authors found no significant benefits in cardiovascular disease from the lifestyle intervention after a follow-up of up to 24 months. This study focuses on the short/long-term benefits of healthy lifestyle intervention in patients from the developed area and offers abundant data , which brings some scientific significance. I listed several major concerns need to be addressed.
Comment 1
The healthy lifestyle intervention was limited to 5 months. Is this long enough for the protective effect to be produced? Within such a short intervention window, the compliance of the patients is important. Is this healthy lifestyle intervention a supervised process? Or just self-regulated by the patients?
Response 1
Thank you for pointing this out. The relatively short duration of the programme indeed may have influenced the effectiveness on cardiovascular and lifestyle behaviour outcomes. Therefore, we added the following to the discussion section, lines 373-375, to address this issue:
Furthermore, the five-month supported programme may have been too short to result in significant changes. However, currently implemented Dutch lifestyle interventions with an intensive treatment period of 6-8 months have been proven effective (Duijzer, Haveman-Nies et al. 2017, Van Rinsum, Gerards et al. 2018). Total follow-up time was 2 years: if participants of the programme would have maintained their lifestyle adaptations, one should expect improved outcomes for those participated in the programme.
For all participants, we hypothesize that people possibly did not receive the right support to achieve their behaviour change goals, as stated in the discussion section, line 399-400.
To clarify the supervision during the programme, we added the following to the methods section, line 113-114:
It consisted of eight group sessions (8-10 persons) and three individual sessions supervised by a lifestyle coach […]
Comment 2
In the end, only 40 patients with cardiovascular risk meaningfully accepted the healthy lifestyle intervention. This small sample size is very likely to limit the power of the study and prevents a possibly true effect from being observed. How do the authors handle this issue?
Response 2
Thank you for this important consideration. We addressed this by measuring the outcomes on a population level, being: those who are offered to participate in the programme. We think that the relatively low number of patients participating in the programme also is an important outcome of our study, showing the difference between modelling studies versus a real-world setting, where implementation of such interventions face various difficulties as described by Berendsen (2015), Van Rinsum (2019), Van der Heiden (2022) and Smit (unpublished). This is outlined in our aims (lines 75-77) , methods (statistical analyses, from line 186) and we also address this in the discussion. We performed a sub-analysis to examine the effects of those actually participated in the programme versus the control group. Our findings support the recently published report on the early effects of the reimbursed lifestyle intervention for people with obesity in The Netherlands, reporting recruitment problems, a drop-out rate of 20% and limited first effects on BMI reduction.
Comment 3
According to the BMI and alcohol use, the overall lifestyle of the enrolled patients was not that hazardous. It’s not surprising due to the health education in the developed area like Netherland. Extra lifestyle intervention may not exert further effects on a prominent level. Did the authors take into account the correct sampled population?
Response 3
Thank you for this comment. The number of participants already meeting the Dutch guidelines for healthy alcohol consumption (1 or less glasses of alcohol per day) is indeed slightly higher than in the general population of The Hague (The Hague: 52%, control group: 61% and intervention group 68%). It is therefore not a representative group of the general population.
We consider the weight of our population representative for a high risk-population in The Netherlands. The mean weight of the Dutch population in 2021 was 78,3 kg, slightly lower than the (normally distributed) mean weight in our population It should be noted that people with pre-existent cardiovascular disease and diabetes mellitus type 2 were not included in this study. Especially people with type 2 diabetes are generally overweight; by excluding them the mean weight of our population may have been relatively lowered in comparison to the full population with high cardiovascular risk.
We added both limitations to the limitation section of the discussion from line 442:
We did not recruit a fully representative sample of the high risk population in our region. Results may therefore not generalisable to other regions [42]. Participating practices and the lifestyle programme were located throughout socio-economic diverse neighbourhoods in an urbanized region, while, compared to the target population, the number of people included in the study with a migrant background or lower education was relatively low. Regarding weight reduction, the mean weight of our population was representative for the target population (Centraal Bureau voor de Statistiek 2021). However, generally overweight people with type 2 diabetes (WHO 2014) were not recruited, in the largest effect of an weight-reduction intervention may have been found. Furthermore, the percentage of smokers in our study was low, therefore the effect of the trial on quitting smoking could only be estimated to a limited extent. This low percentage is in line with the percentage of smokers in The Hague aged>65 years or with a higher educational level, thus representative for the population that was reached by this study [42]. However, higher percentages of smokers are known to be among people within groups with a migration background and with lower educational levels, who were not fully reached. Lastly, the number of participants already meeting the Dutch guidelines for healthy alcohol consumption (1 or less glasses of alcohol per day) is slightly higher than in the general population of The Hague (The Hague: 52%, control group: 61% and intervention group 68%), thus not meeting the target high-risk population, namely those with high alcohol consumption.
Reviewer 3 Report
The manuscript is interesting to the scientific community. Please find my remarks below:
1. I suggest the authors to formulate key research questions. This should ideally be addressed in the introductory section
2. Please highlight your main contributions in the introductory section.
3. Please provide an analysis of data quality including the percentage of outliers that may exist in the data. How were the outliers handled?
Author Response
Reviewer 3
We highly appreciate the time and effort that the reviewer has dedicated to provide valuable feedback on our manuscript. Thank you for giving us the opportunity to submit a revised draft of the manuscript. Please see below a point-by-point response to the reviewers’ comments and concerns.
Comment 1
I suggest the authors to formulate key research questions. This should ideally be addressed in the introductory section
Response 1
Thank you for this suggestion. We added the research questions to the introduction section: line 78-82:
[…] We examined the effect of offering the programme on weight, systolic blood pressure (SBP), cholesterol levels, diet quality, physical activity, smoking and alcohol consumption in the target population. Furthermore, we examined goalsetting on lifestyle behaviour and self-reported achievement of goals. To examine the cost-benefit of implementing the programme we performed a cost-utility analysis.
Comment 2
Please highlight your main contributions in the introductory section.
Response 2
Thank you for this suggestion. In response to your suggestion we added the following to the introduction section, line 82-84:
[…] With this study, we provide new insights on the effects of offering a lifestyle intervention in a real-life primary care setting.
Comment 3
Please provide an analysis of data quality including the percentage of outliers that may exist in the data. How were the outliers handled?
Response 3
Outliers were handled by using a mixed model for the analysis: by taking the individual participant as a random effect and correcting for baseline values, the impact of outliers was limited. Before analysis, all outcome variables were assessed on normality by histograms. In response to the reviewer we performed an outlier analysis, which is attached to this response.

Round 2
Reviewer 1 Report
The manuscript can be accepted this form.
Author Response
We thank the reviewer for his/her positive opinion on our manuscript.
Reviewer 2 Report
This manuscript describes a clinical trial in which 787 patients with or without cardiovascular risks were enrolled, and lifestyle intervention were given to 40 patients with cardiovascular risks. The authors found no significant benefits in cardiovascular disease from the lifestyle intervention from a follow-up of up to 24 months. This study focuses on the short/long-term benefits of healthy lifestyle intervention in patients from the developed area, which brings some scientific significance. The authors responded well to my questions and made some revisions. After explanations, the conclusion is less vulnerable. Overall, the manuscript offers abundant data and the tables and figures are logically organized.
Author Response
We thank the reviewer for his/her positive feedback on our revisions.
Reviewer 3 Report
There appears to be an issue with the reference section. The majority of the references have vanished. Kindly correct it.
Author Response
We thank the reviewer for his/her valuable comment. The configuration of the references between version 1 of the manuscript and the revised version unfortunately failed. We corrected this in the current version.